# Weakly-supervised Discovery of Visual Pattern Configurations

**Hyun Oh Song**      **Yong Jae Lee**[*]      **Stefanie Jegelka**      **Trevor Darrell**

University of California, Berkeley      [*]University of California, Davis

## Abstract

The prominence of weakly labeled data gives rise to a growing demand for object detection methods that can cope with minimal supervision. We propose an approach that automatically identifies discriminative configurations of visual patterns that are characteristic of a given object class. We formulate the problem as a constrained submodular optimization problem and demonstrate the benefits of the discovered configurations in remedying mislocalizations and finding informative positive and negative training examples. Together, these lead to state-of-the-art weakly-supervised detection results on the challenging PASCAL VOC dataset.

## 1   Introduction

The growing amount of sparsely and noisily labeled image data demands robust detection methods that can cope with a minimal amount of supervision. A prominent example of this scenario is the abundant availability of labels at the image level (i.e., whether a certain object is present or absent in the image); detailed annotations of the exact location of the object are tedious and expensive and, consequently, scarce. Learning methods that can handle image-level labels circumvent the need for such detailed annotations and therefore have the potential to effectively use the vast textually annotated visual data available on the Web. Moreover, if the detailed annotations happen to be noisy or erroneous, such weakly supervised methods can even be more robust than fully supervised ones.

Motivated by these developments, recent work has explored learning methods that decreasingly rely on strong supervision. Early ideas for weakly supervised detection [11, 32] paved the way by successfully learning part-based object models, albeit on simple object-centric datasets (e.g., Caltech-101). Since then, a number of approaches [21, 26, 29] have aimed at learning models from more realistic and challenging data sets that feature large intra-category appearance variations and background clutter. These approaches typically generate multiple candidate regions and retain the ones that occur most frequently in the positively-labeled images. However, due to intra-category variations and deformations, the identified (single) patches often correspond to only a part of the object, such as a human face instead of the entire body. Such mislocalizations are a frequent problem for weakly supervised detection methods.

Mislocalization and too large or too small bounding boxes are problematic in two respects. First, detection is commonly phrased as multiple instance learning (MIL) and solved by non-convex optimization methods that alternatively guess the location of the objects as positive examples (since the true location is unknown) and train a detector based on those guesses. This procedure is heavily affected by the initial localizations. Second, the detector is often trained in stages; in each stage one adds informative "hard" negative examples to the training data. If we are not given accurate true object localizations in the training data, these hard examples must be derived from the detections inferred in earlier rounds. The higher the accuracy of the initial localizations, the more informative is the augmented training data – and this is key to the accuracy of the final learned model.

In this work, we address the issue of mislocalizations by identifying characteristic, discriminative *configurations* of multiple patches (rather than a single one). This part-based approach is motivated

by the observation that automatically discovered single "discriminative" patches often correspond to object parts. In addition, while background patches (e.g., of water or sky) can also occur throughout the positive images, they will re-occur in arbitrary rather than "typical" configurations. We develop an effective method that takes as input a set of images with labels of the form "the object is present/absent", and automatically identifies characteristic part configurations of the given object.

To identify such configurations, we use two main criteria. First, useful patches are *discriminative*, i.e., they occur in many positively-labeled images, and rarely in the negatively labeled ones. To identify such patches, we use a discriminative covering formulation similar to [29]. Second, the patches should represent *different* parts, i.e., they may be close but should not overlap too much. In covering formulations, one may rule out overlaps by saying that for two overlapping regions, one "covers" the other, i.e., they are treated as identical and picking one is as good as picking both. But identity is a transitive relation, and the density of possible regions in detection would imply that all regions are identical, strongly discouraging the selection of more than one part per image. Partial covers face the problem of scale invariance. Hence, we instead formulate an independence constraint. This second criterion ensures that we select regions that may be close but are non-redundant and sufficiently non-overlapping. We show that this constrained selection problem corresponds to maximizing a submodular function subject to a matroid intersection constraint, which leads to approximation algorithms with theoretical worst-case bounds. Given candidate parts identified by these two criteria, we effectively find frequently co-occurring configurations that take into account relative position, scale, and viewpoint.

We demonstrate multiple benefits of the discovered configurations. First, we observe that configurations of patches can produce more accurate spatial coverage of the full object, especially when the most discriminative pattern corresponds to an object part. Second, any overlapping region between co-occurring visual patterns is likely to cover a part (but not the full) of the object of interest. Thus, they can be used to generate mis-localized positives as informative hard negatives for training (see white boxes in Figure 3), which can further reduce localization errors at test time.

In short, our main contribution is a weakly-supervised object detection method that automatically discovers frequent configurations of discriminative visual patterns to train robust object detectors. In our experiments on the challenging PASCAL VOC dataset, we find the inclusion of our discriminative, automatically detected configurations to outperform all existing state-of-the-art methods.

## 2 Related work

**Weakly-supervised object detection.** Object detectors have commonly been trained in a fully-supervised manner, using tight bounding box annotations that cover the object of interest (e.g., [10]). To reduce laborious bounding box annotation costs, recent weakly-supervised approaches [3, 4, 11, 21, 26, 29, 32] use image-level object-presence labels with no information on object location.

Early efforts [11, 32] focused on simple datasets that have a single prominent object in each image (e.g., Caltech-101). More recent approaches [21, 26, 29] work with the more challenging PASCAL dataset that contains multiple objects in each image and large intra-category appearance variations. Of these, Song et al. [29] achieve state-of-the-art results by finding discriminative image patches that occur frequently in the positive images but rarely in the negative images, using deep Convolutional Neural Network (CNN) features [17] and a submodular cover formulation. We build on their approach to identify discriminative patches. But, contrary to [29] which assumes patches to contain entire objects, we assume patches to contain either full objects or merely object *parts*, and automatically piece together those patches to produce better full-object estimates. To this end, we change the covering formulation and identify patches that are both representative and explicitly mutually different. This leads to more robust object estimates and further allows our system to intelligently select "hard negatives" (mislocalized objects), both of which improve detection performance.

**Visual data mining.** Existing approaches discover high-level object categories [14, 7, 28], mid-level patches [5, 16, 24], or low-level foreground features [18] by grouping similar visual patterns (i.e., images, patches, or contours) according to their texture, color, shape, etc. Recent methods [5, 16] use weakly-supervised labels to discover discriminative visual patterns. We use related ideas, but formulate the problem as a submodular optimization over matroids, which leads to approximation algorithms with theoretical worst-case guarantees. Covering formulations have also been used in

[1, 2], but *after* running a trained object detector. An alternative discriminative approach is to use spectral methods [34].

**Modeling co-occurring visual patterns.** It is known that modeling the spatial and geometric relationship between co-occurring visual patterns (objects or object-parts) often improves visual recognition performance [8, 18, 10, 11, 19, 23, 27, 24, 32, 33]. Co-occurring patterns are usually represented as doublets [24], higher-order constellations [11, 32] or star-shaped models [10]. Among these, our work is most inspired by [11, 32], which learn part-based models with weak supervision. We use more informative deep CNN features and a different formulation, and show results on more difficult datasets. Our work is also related to [19], which discovers high-level object compositions ("visual phrases" [8]), but with ground-truth bounding box annotations. In contrast, we aim to discover part compositions to represent full objects and do so with less supervision.

## 3    Approach

Our goal is to find a discriminative set of patches that co-occur in the same configuration in many positively-labeled images. We address this goal in two steps. First, we find a set of patches that are discriminative; i.e., they occur frequently in positive images and rarely in negative images. Second, we efficiently find co-occurring configurations of pairs of such patches. Our approach easily extends beyond pairs; for simplicity and to retain configurations that occur frequently enough, we here restrict ourselves to pairs.

**Discriminative candidate patches.** For identifying discriminative patches, we begin with a construction similar to that of Song et al. [29]. Let $\mathcal{P}$ be the set of positively-labeled images. Each image $I$ contains candidate boxes $\{b_{I,1}, \ldots, b_{I,m}\}$ found via selective search [30]. For each $b_{I,i}$, we find its closest matching neighbor $b_{I',j}$ in each other image $I'$ (regardless of the image label). The $K$ closest of those neighbors form the neighborhood $\mathcal{N}(b_{I,i})$; the remaining ones are discarded.

Discriminative patches have neighborhoods mainly within images in $\mathcal{P}$, i.e., if $\mathcal{B}(\mathcal{P})$ is the set of all patches from images in $\mathcal{P}$, then $|\mathcal{N}(b) \cap \mathcal{B}(\mathcal{P})| \approx K$. To identify a small, diverse and representative set of such patches, like [29], we construct a bipartite graph $\mathcal{G} = (\mathcal{U}, \mathcal{V}, \mathcal{E})$, where both $\mathcal{U}$ and $\mathcal{V}$ contain copies of $\mathcal{B}(\mathcal{P})$. Each patch $b \in \mathcal{V}$ is connected to the copy of its nearest neighbors in $\mathcal{U}$ (i.e., $\mathcal{N}(b) \cap \mathcal{B}(\mathcal{P})$). These will be $K$ or fewer, depending on whether the $K$ nearest neighbors of $b$ occur in $\mathcal{B}(\mathcal{P})$ or in negatively-labeled images. The most representative patches maximize the covering function

$$F(S) = |\Gamma(S)|, \tag{1}$$

where $\Gamma(S) = \{u \in \mathcal{U} \mid (b, u) \in \mathcal{E} \text{ for some } b \in S\} \subseteq \mathcal{U}$ is the neighborhood of $S \subseteq \mathcal{V}$ in the bipartite graph. Figure 1 shows a cartoon illustration. The function $F$ is monotone and submodular, and the $C$ maximizing elements (for a given $C$) can be selected greedily [20].

However, if we aim to find part configurations, we must select multiple, jointly informative patches per image. Patches selected to merely maximize coverage can still be redundant, since the most frequently occurring ones are often highly overlapping. A straightforward modification would be to treat highly overlapping patches as identical. This identification would still admit a submodular cover model as in Equation (1). But, in our case, the candidate patches are very densely packed in the image, and, by transitivity, we would have to make all of them identical. In consequence, this would completely rule out the selection of more than one patch in an image and thereby prohibit the discovery of any co-occurring configurations.

Instead, we directly constrain our selection such that no two patches $b, b' \in \mathcal{V}$ can be picked whose neighborhoods overlap by more than a fraction $\theta$. By overlap, we mean that the patches in the neighborhoods of $b, b'$ overlap significantly (they need not be identical). This notion of diversity is reminiscent of NMS and similar to that in [5], but we here phrase and analyze it as a constrained submodular optimization problem. Our constraint can be expressed in terms of a different graph $\mathcal{G}_C = (\mathcal{V}, \mathcal{E}_C)$ with nodes $\mathcal{V}$. In $\mathcal{G}_C$, there is an edge between $b$ and $b'$ if their neighborhoods overlap prohibitively, as illustrated in Figure 1. Our family of feasible solutions is

$$\mathcal{M} = \{S \subseteq V \mid \forall b, b' \in S \text{ there is no edge } (b, b') \in \mathcal{E}_C\}. \tag{2}$$

In other words, $\mathcal{M}$ is the family of all independent sets in $\mathcal{G}_C$. We aim to maximize

$$\max_{S \subseteq \mathcal{V}} F(S) \quad \text{s.t. } S \in \mathcal{M}. \tag{3}$$

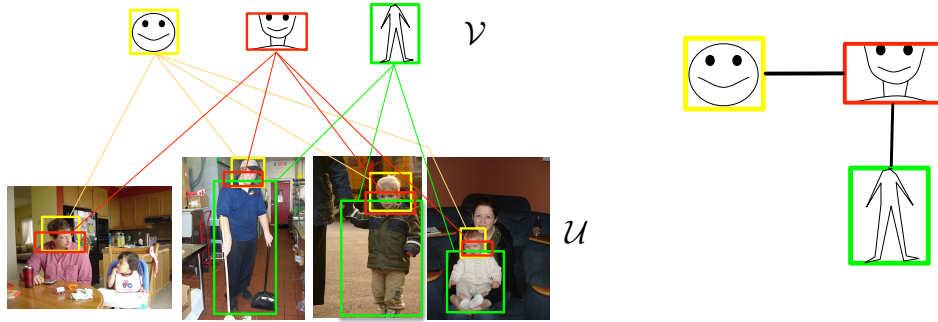

Figure 1: Left: bipartite graph $\mathcal{G}$ that defines the utility function $F$ and identifies discriminative patches; right: graph $\mathcal{G}_C$ that defines the diversifying independence constraints $\mathcal{M}$. We may pick $C_1$ (yellow) and $C_3$ (green) together, but not $C_2$ (red) with any of those.

This problem is NP-hard. We solve it approximately via the following greedy algorithm. Begin with $S^0 = \emptyset$, and, in iteration $t$, add $b \in \mathrm{argmax}_{b \in \mathcal{V} \setminus S} |\Gamma(b) \setminus \Gamma(S^{t-1})|$. As we add $b$, we delete all of $b$'s neighbors in $\mathcal{G}_C$ from $\mathcal{V}$. We continue until $\mathcal{V} = \emptyset$. If the neighborhoods of any $b, b'$ are disjoint but contain overlapping elements ($\Gamma(b) \cap \Gamma(b') = \emptyset$ but there exist $u \in \Gamma(b)$ and $u' \in \Gamma(b')$ that overlap), then this algorithm amounts to the following simplified scheme: we first sort all $b \in \mathcal{V}$ in non-increasing order by their degree $\Gamma(b)$, i.e., their number of neighbors in $\mathcal{B}(\mathcal{P})$, and visit them in this order. We always add the currently highest $b$ in the list to $S$, then delete it from the list, and with it all its immediate (overlapping) neighbors in $\mathcal{G}_C$. The following lemma states an approximation factor for the greedy algorithm, where $\Delta$ is the maximum degree of any node in $\mathcal{G}_C$.

**Lemma 1.** *The solution $S_g$ returned by the greedy algorithm is a $1/(\Delta + 2)$ approximation for Problem (2): $F(S_g) \geq \frac{1}{\Delta+2} F(S^*)$. If $\Gamma(b) \cap \Gamma(b') = \emptyset$ for all $b, b' \in \mathcal{V}$, then the worst-case approximation factor is $1/(\Delta + 1)$.*

The proof relies on phrasing $\mathcal{M}$ as an intersection of matroids.

**Definition 1** (Matroid). *A matroid $(\mathcal{V}, \mathcal{I}_k)$ consists of a ground set $\mathcal{V}$ and a family $\mathcal{I}_k \subseteq 2^{\mathcal{V}}$ of "independent sets" that satisfy three axioms: (1) $\emptyset \in \mathcal{I}_k$; (2) downward closedness: if $S \in \mathcal{I}_k$ then $T \in \mathcal{I}_k$ for all $T \subseteq S$; and (3) the exchange property: if $S, T \in \mathcal{I}_k$ and $|S| < |T|$, then there is an element $v \in T \setminus S$ such that $S \cup \{v\} \in \mathcal{I}_k$.*

*Proof. (Lemma 1)* We will argue that Problem (2) is the problem of maximizing a monotone submodular function subject to the constraint that the solution lies in the intersection of $\Delta + 1$ matroids. With this insight, the approximation factor of the greedy algorithm for submodular $F$ follows from [12] and that for non-intersecting $\Gamma(b)$ from [15], since in the latter case the problem is that of finding a maximum weight vector in the intersection of $\Delta + 1$ matroids.

It remains to argue that $\mathcal{M}$ is an intersection of matroids. Our matroids will be partition matroids (over the ground set $\mathcal{V}$) whose independent sets are of the form $\mathcal{I}_k = \{S \mid |S \cap e| \leq 1, \text{ for all } e \in E_k\}$. To define those, we partition the edges in $\mathcal{G}_C$ into disjoint sets $E_k$, i.e., no two edges in $E_k$ share a common node. The $E_k$ can be found by an edge coloring – one $E_k$ and $\mathcal{I}_k$ for each color $k$. By Vizing's theorem [31], we need at most $\Delta + 1$ colors. The matroid $\mathcal{I}_k$ demands that for each edge $e \in E_k$, we may only select one of its adjacent nodes. All matroids together say that for any edge $e \in \mathcal{E}$, we may only select one of the adjacent nodes, and that is the constraint in Equation (2), i.e. $\mathcal{M} = \bigcap_{k=1}^{\Delta+1} \mathcal{I}_k$. We do not ever need to explicitly compute $E_k$ and $\mathcal{I}_k$; all we need to do is check membership in the intersection, and this is equivalent to checking whether a set $S$ is an independent set in $\mathcal{G}_C$, which is achieved implicitly via the deletions in the algorithm. $\qquad\square$

From the constrained greedy algorithm, we obtain a set $S \subset \mathcal{V}$ of discriminative patches. Together with its neighborhood $\Gamma(b)$, each patch $b \in \mathcal{V}$ forms a representative cluster. Figure 2 shows some example patches derived from the labels "aeroplane" and "motorbike". The discovered patches intuitively look like "parts" of the objects, and are frequent but sufficiently different.

**Finding frequent configurations.** The next step is to find frequent configurations of co-occurring clusters, e.g., the head patch of a person on top of the torso patch, or a bicycle with visible wheels.

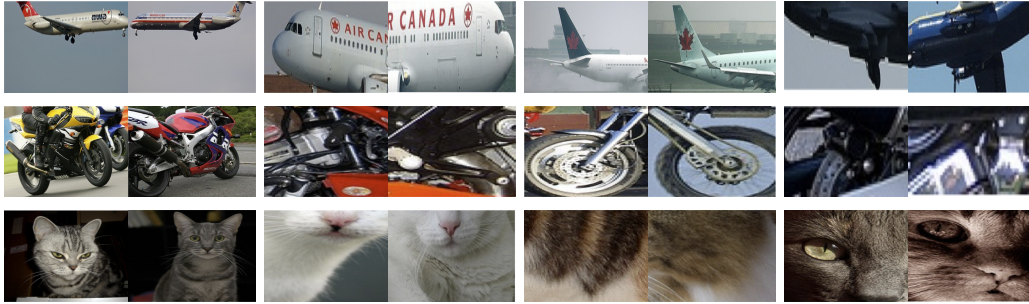

Figure 2: Examples of discovered patch "clusters" for aeroplane, motorbike, and cat. The discovered patches intuitively look like object parts, and are frequent but sufficiently different.

A "configuration" consists of patches from two clusters $C_i, C_j$, their relative location, and their viewpoint and scale. In practice, we give preference to pairs that by themselves are very relevant and maximize a weighted combination of co-occurrence count and coverage $\max\{\Gamma(C_i), \Gamma(C_j)\}$.

All possible configurations of all pairs of patches amount to too many to explicitly write down and count. Instead, we follow an efficient procedure for finding frequent configurations. Our approach is inspired by [19], but does not require any supervision. We first find configurations that occur in at least two images. To do so, we consider each pair of images $I_1, I_2$ that have at least two co-occurring clusters. For each correspondence of cluster patches across the images, we find a corresponding transform operation (translation, scale, viewpoint change). This results in a point in a 4D transform space, for each cluster correspondence. We quantize this space into $B$ bins. Our candidate configurations will be pairs of cluster correspondences $((b_{I_1,1}, b_{I_2,1}), (b_{I_1,2}, b_{I_2,2})) \in (C_i \times C_i) \times (C_j \times C_j)$ that fall in the same bin, i.e., share the same transform and have the same relative location. Between a given pair of images, there can be multiple such pairs of correspondences. We keep track of those via a multi-graph $\mathcal{G}_P = (\mathcal{P}, \mathcal{E}_P)$ that has a node for each image $I \in \mathcal{P}$. For each correspondence $((b_{I_1,1}, b_{I_2,1}), (b_{I_1,2}, b_{I_2,2}))$, we draw an edge $(I_1, I_2)$ and label it by the clusters $C_i, C_j$ and the common relative position. As a result, there can be multiple edges $(I_1, I_j)$ in $\mathcal{G}_P$ with different edge labels.

The most frequently occurring configuration can now be read out by finding the largest connected component in $\mathcal{G}_P$ induced by retaining only edges with the same label. We use the largest component(s) as the characteristic configurations for a given image label (object class). If the component is very small, then there is not enough information to determine co-occurrences, and we simply use the most frequent single cluster. The final single "correct" localization will be the smallest bounding box that contains the full configuration.

**Discovering mislocalized hard negatives.** Discovering frequent configurations can not only lead to better localization estimates of the full object, but they can also be used to generate mislocalized estimates as "hard negatives" when training the object detector. We exploit this idea as follows. Let $b_1, b_2$ be a discovered configuration within a given image. These patches typically constitute co-occurring parts or a part and the full object. Our foreground estimate is the smallest box that includes both $b_1$ and $b_2$. Hence, any region within the foreground estimate that does not overlap simultaneously with both $b_1$ and $b_2$ will capture only a *fragment* of the foreground object. We extract the four largest such rectangular regions (see white boxes in Figure 3) as hard negative examples.

Specifically, we parameterize any rectangular region with $[x^l, x^r, y^t, y^b]$, i.e., its $x$-left, $x$-right, $y$-top, and $y$-bottom coordinate values. Let the bounding box of $b_i$ ($i = 1, 2$) be $[x_i^l, x_i^r, y_i^t, y_i^b]$, the foreground estimate be $[x_f^l, x_f^r, y_f^t, y_f^b]$, and let $x^l = \max(x_1^l, x_2^l)$, $x^r = \min(x_1^r, x_2^r)$, $y^t = \max(y_1^t, y_2^t)$, $y^b = \min(y_1^b, y_2^b)$. We generate four hard negatives: $[x_f^l, x^l, y_f^b, y_f^t]$, $[x^r, x_f^r, y_f^b, y_f^t]$, $[x_f^l, x_f^r, y_f^t, y^t]$, $[x_f^l, x_f^r, y^b, y_f^b]$. If either $b_1$ or $b_2$ is very small in size relative to the foreground, the resulting hard negatives can have high overlap with the foreground, which will introduce undesirable noise (false negatives) when training the detector. Thus, we shrink any hard negative that overlaps with the foreground estimate by more than 50%, until its overlap is 50% (we adjust the boundary that does not coincide with any of the foreground estimation boundaries).

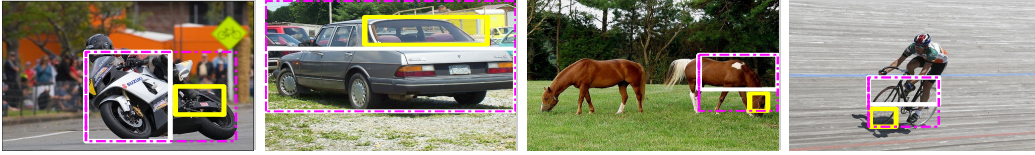

Figure 3: Automatically discovered foreground estimation box (magenta), hard negative (white), and the patch (yellow) that induced the hard negative. Note that we are only showing the largest one out of (up to) four hard negatives per image.

Note that simply taking arbitrary rectangular regions that overlap with the foreground estimation box by some threshold will not always generate useful hard negatives (as we show in the experiments). If the overlap threshold is too low, the selected regions will be uninformative, and if the overlap threshold is too high, the selected regions will cover too much of the foreground. Our approach selects informative hard negatives more robustly by ruling out the overlapping region between the configuration patches, which is very likely be part of the foreground object but not the full object.

**Mining positives and training the detector.** While the discovered configurations typically lead to better foreground localization, their absolute count can be relatively low compared to the total number of positive images. This is due to inaccuracies in the initial patch discovery stage: for a frequent configuration to be discovered, both of its patches must be found accurately. Thus, we also mine additional positives from the set of remaining positive images $\mathcal{P}'$ that did not produce any of the discovered configurations.

To do so, we train an initial object detector, using the foreground estimates derived from our discovered configurations as positive examples, and the corresponding discovered hard negative regions as negatives. In addition, we mine negative examples in negative images as in [10]. We run the detector on all selective search regions in $\mathcal{P}'$ and retain the region in each image with the highest detection score as an additional positive training example. Our final detector is trained on this augmented training data, and iteratively improved by latent SVM (LSVM) updates (see [10, 29] for details).

## 4 Experiments

In this section, we analyze: (1) detection performance of the models trained with the discovered configurations, and (2) impact of the discovered hard negatives on detection performance.

**Implementation details.** We employ a recent region based detection framework [13, 29] and use the same fc7 features from the CNN model [6] on region proposals [30] throughout the experiments. For discriminative patch discovery, we use $K = |\mathcal{P}|/2, \theta = K/20$. For correspondence detection, we discretize the 4D transform space of $\{x$: relative horizontal shift, $y$: relative vertical shift, $s$: relative scale, $p$: relative aspect ratio$\}$ with $\Delta x = 30 \ px, \Delta y = 30 \ px, \Delta s = 1 \ px/px, \Delta p = 1 \ px/px$. We chose this binning scheme by examining a few qualitative examples so that scale and aspect ratio agreement between the two paired instances are more strict, while their translation agreement is more loose, in order to handle deformable objects. More details regarding the transform space binning can be found in [22].

**Discovered configurations.** Figure 5 shows the discovered configurations (solid green and yellow boxes) and foreground estimates (dashed magenta boxes) that have high degree in graph $\mathcal{G}_P$ for all 20 classes in the PASCAL dataset. Our method consistently finds meaningful combinations such as a wheel and body of bicycles, face and torso of people, locomotive basement and upper body parts of trains/buses, and window and body frame of cars. Some failures include cases where the algorithm latches onto different objects co-occurring in consistent configurations such as the lamp and sofa combination (right column, second row from the bottom in Figure 5).

**Weakly-supervised object detection.** Following the evaluation protocol of the PASCAL VOC dataset, we report detection results on the PASCAL *test* set using detection average precision. For a direct comparison with the state-of-the-art weakly-supervised object detection method [29], we do not use the extra instance level annotations such as *pose, difficult, truncated* and restrict the supervision to the image-level object presence annotations. Table 1 compares our detection results against two baseline methods [25, 29] on the full dataset. Our method improves detection performance on 15 of the 20 classes. It is worth noting that our method yields significant improvement on the person

| | aero | bike | bird | boat | btl | bus | car | cat | chr | cow | tble | dog | horse | mbk | pson | plnt | shp | sofa | train | tv | mAP |
|---|---|---|---|---|---|---|---|---|---|---|---|---|---|---|---|---|---|---|---|---|---|
| [25] | 13.4 | 44.0 | 3.1 | 3.1 | 0.0 | 31.2 | 43.9 | 7.1 | 0.1 | 9.3 | 9.9 | 1.5 | **29.4** | 38.3 | 4.6 | 0.1 | 0.4 | 3.8 | 34.2 | 0.0 | 13.9 |
| [29] | 27.6 | 41.9 | 19.7 | 9.1 | 10.4 | 35.8 | 39.1 | **33.6** | 0.6 | 20.9 | 10.0 | **27.7** | 29.4 | 39.2 | 9.1 | **19.3** | 20.5 | **17.1** | 35.6 | 7.1 | 22.7 |
| ours[1] | 31.9 | 47.0 | 21.9 | 8.7 | 4.9 | 34.4 | 41.8 | 25.6 | 0.3 | 19.5 | **14.2** | 23.0 | 27.8 | 38.7 | **21.2** | 17.6 | **26.9** | 12.8 | **40.1** | 9.2 | 23.4 |
| ours[2] | **36.3** | **47.6** | **23.3** | **12.3** | **11.1** | **36.0** | **46.6** | 25.4 | **0.7** | **23.5** | 12.5 | 23.5 | 27.9 | **40.9** | 14.8 | 19.2 | 24.2 | **17.1** | 37.7 | **11.6** | **24.6** |

Table 1: Detection average precision (%) on full PASCAL VOC 2007 test set. ours[1]: before latent updates. ours[2]: after latent updates

| | w/o hard negatives | neighboring hard negatives | discovered hard negatives |
|---|---|---|---|
| ours + SVM | 22.5 | 22.2 | **23.4** |
| ours + LSVM | 23.7 | 23.9 | **24.6** |

Table 2: Effect of our hard negative examples on full PASCAL VOC 2007 test set.

class, which is arguably the most important category in the PASCAL dataset. Figure 4 shows some example high scoring detections on the test set. Our method produces more complete detections since it is trained on better localized instances of the object-of-interest.

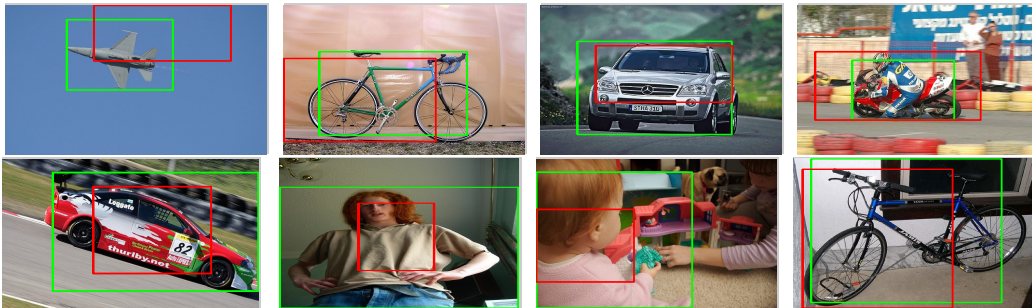

Figure 4: Example detections on test set. Green: our method, red: [29]

**Impact of discovered hard negatives.** To analyze the effect of our discovered hard negatives, we compare to two baselines: (1) not adding any negative examples from positives images, and (2) adding image regions around the foreground estimate, as conventionally implemented in fully supervised object detection algorithms [9, 13]. For the latter, we use the criterion from [13], where all image regions in positive images with overlap score (intersection over union with respect to any foreground region) less than 0.3 are used as "neighboring" negative image regions on positive images. Table 2 shows the effect of our hard negative examples on detection mean average precision for all classes (mAP). We also added neighboring negative examples to [29], but this decreases its mAP from 20.3% to 20.2% (before latent updates) and from 22.7% to 21.8% (after latent updates). These experiments show that adding neighboring negative regions does not lead to noticeable improvement over not adding any negative regions from positive images, while adding our automatically discovered hard negative regions improves detection performance more substantially.

**Conclusion.** We developed a weakly-supervised object detection method that discovers frequent configurations of discriminative visual patterns. We showed that the discovered configurations provide more accurate spatial coverage of the full object and provide a way to generate useful hard negatives. Together, these lead to state-of-the-art weakly-supervised detection results on the challenging PASCAL VOC dataset.

**Acknowledgement.** This work was supported in part by DARPA's MSEE and SMISC programs, by NSF awards IIS-1427425, IIS-1212798, IIS-1116411, and by support from Toyota.

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
