[Reviews · NeurIPS 2014]

Submitted by Assigned_Reviewer_5

Paper describes a method to identify image patches that are (a) diagnostic of particular objects (b) not particularly redundant and (c) cover well the collection of diagnostic patches. The method applies to the weakly supervised case, where images are known to contain the object(s) of interest, but the location of these objects is not known. This is a very well studied topic. Once these patches have been identified, related pairs are found by a mining process. The method produces patches that outperform those produced by other constructions for this case on a reasonable choice of dataset.

The method is nicely formalized as a matroid intersection problem; authors perhaps underplay the significance of the delta term, as in practical image problems I'd expect delta to be big, and hence the approximation factor to be small.

The method for finding relations makes sense; it's essentially a hough transform lightly disguised as a graph algorithm, and authors might mention this. Hough transforms have standard weaknesses (peaks occur by accident as a result of noise) and
authors might mention why these aren't a problem. The account of parametrization in
ll242 "(translation, rescale, viewpoint change)" is sloppy - what do you mean by viewpoint change? why does it have only
one parameter? could this be rotation? This should be fixed.

Authors mine hard negatives quite aggressively to improve box localization. This seems to work. I'd be interested to know if
there were limits to this process --- at some level, box localization *should* be hard, because the boxes aren't canonical in any sense. Could one end up overtraining the boxes? and so suffer from the amount of noise created by human annotators?
Summary: Paper covers fairly well trodden ground, but supplies a construction of interest, with some minor problems. Paper should likely be published.

Submitted by Assigned_Reviewer_13

The goal of this paper is to learn an object detector that produces tight bounding boxes around objects given training data that contains only weak annotations of presence/absence of objects without bounding box supervision. The paper develops an initialization for the latent SVM detector of Felzenszwalb et al. The paper first obtains many candidate part bounding boxes in all training images using selective search [28]. Then the method proceeds by selecting a set of distinctive object parts that occur (are matched) on the positive images and not on the negative data, but do not significantly overlap each other. This is formulated as a constrained discrete optimization problem and approximately solved using a greedy algorithm. Finally, to better localize the object in the image the paper selects pairs of discriminative parts that tend to co-occurr in a consistent mutual geometric configuration (position, scale and aspect ratio). Bounding boxes obtained from such consistent pairs are then used to train the LSVM object detector.

Novelty:
The work is somewhat incremental w.r.t. (Song et al., ICML'14 [27]).
The main differences are:
1. finding multiple non-overlapping discriminative parts per image. This is in contrast to a single part per image found in [Song et al. ICML'14] and seems as a step in the right direction.
2. finding pairs of parts that occur in a certain configuration that is
based on the algorithm of (Li et al., CVPR 2012 [17]) and involves voting in a 4D part-to-part transformation space.
Pairs of co-occurring parts are used to better localize the object in the image.

While the paper is incremental, the novelty seems sufficient for publication.

Experiments:
- Results are shown on the Pascal VOC 2007 detection dataset.
Results on the test data slightly outperform the method of Song et al. (mAP 24.6 vs. mAP 22.7). The paper also experimentally demonstrates the (slight) improvement of the proposed hard-negative mining procedure that extracts (hard) negative bounding boxes from the positive images.

- The paper also shows qualitative (fig.5) demonstrating the output object localization for several object classes. The qualitative results seem reasonable.

Impact:
While this is a solid work, given the somewhat incremental nature w.r.t. Song et al. its impact is probably going to be limited. In general, the practical impact of weakly supervised learning methods is going to be determined by the scalabality of the method to large datasets with many images and many classes. What is the complexity of the algorithm? How does the algorithm scale to many images and many labels?

Detailed questions:
- L296: It seems that each iteration adds only one additional positive example.
How many iterations are performed? How is the algorithm stopped?

- The paper claims that the algorithm easily extends beyond pairs of parts to three or more parts. This is not clear to me. The authors should elaborate on this in the rebuttal.

- The authors should discuss the scalability of the proposed method to large datasets and many object categories.
Summary: This is a solid paper that addresses the weakly supervised object detection problem.
The experiments demonstrate small but convincing improvement over state-of-the-art on the difficult Pascal VOC 2007 image dataset. On the downside, the paper is only incremental w.r.t. [Song et al. ICML'14].

Submitted by Assigned_Reviewer_31

The paper proposes a method for discovering configurations of recurring image patches from weakly labelled data, i.e. the label is only at the image level. Detectors trained on the discovered configurations perform better than current weakly supervised methods trained on single recurring patches (on PASCAL VOC 2007).

The novelty lies in:
1) The modification of a method for finding clusters of recurring patches [27], by enforcing that patches in different clusters do not overlap. A worst-case bound for this greedy algorithm is provided.
2) A method to find co-occurring configurations of 2 patches from different clusters, enforcing consistent relative location, viewpoint and scale.
3) Generation of hard negatives by using only portions of one part in the configuration and not the other.

Pros:
- The concept that the discovered recurring patches cannot overlap is interesting. Overlapping patches tend to be associated to the same object part, and would prevent the discovery of meaningful configurations
- The paper is quite clear
- The method is technically sound and so is the evaluation

Cons:
- The method is incremental, as it extends [27] to handle configurations of two recurring patches instead of individual ones. It would be much stronger if it could handle more than 2 patches, or even adaptively discover the number of patches in the configuration.
- The improvement in performance over [27] is small (2%), and it seems that half of it is due to the new hard negatives. It would be interesting to add these hard negatives to [27], to validate that they are beneficial only when training on configurations of patches. Obviously the part decomposition is not available in [27], but hard negatives could be generated by using random subwindows with less than 50% overlap.
The authors clarified this point in their rebuttal.

A few mistakes/ambiguities, and other smaller issues:
Line 64: the second “may” should be “but”
Line 95: the authors might consider citing Deselaers ECCV 2010, as it was the work that started to tackle weakly supervised object class localization on the challenging PASCAL dataset.
Line 134: is K a set of patches or an integer? If the latter, line 136 should have the cardinality of the intersection set equals to K. Clarified in the rebuttal.
Line 214: A small title to start this paragraph would help show that there is a transition from the proof to the sectiono on finding configurations.
Line 254-255: this is not clear. What does "single cluster" mean? Just one patch instead of a configuration?
Summary: The paper is technically sound and reasonably clear, with a few minor mistakes. The method is mostly incremental, but with a few interesting ideas, and I lean towards acceptance.
Author Feedback
Author rebuttal: We thank the reviewers for their time and effort. We especially appreciate the encouraging comments ("solid paper sufficient for publication" - R1, "lean towards acceptance" - R2, "should likely be published" - R3).

We'll first address common questions and then answer specific questions.

Groups of more than 2 patches as pattern configurations (R1, R2):
- The same method works for multiple patches too: In the discovery step, we keep track of all the groups in a hash table with hash key of cluster indices and the relative locations in transform space. Configurations correspond to connected components in the multigraph \mathcal{G}_P (defined in L247), and component size indicates frequency. While this is possible, the larger the configuration the more data is needed: on our data, the larger/more reliable components correspond to pairs, e.g. the occurrence counts for pairs vs triplets vs quadraplets are on average 422.4 vs 15.6 vs 1.3 per category. Our results demonstrate that good improvements are possible even with pairs, and correct object boundaries are covered to a substantial extent (see Fig. 5). Since our goal is less of semantically meaningful representation, and rather that of reliably detecting full objects, pairs serve our goal well. With sufficient evidence (frequency), higher order patterns could lead to further improvement. But patterns with rare occurrence would not be suitable to derive a sufficient number of foreground estimates for training object detectors, and they may not even be necessary.

R1:

- We chose PASCAL2007 to make quantitative comparisons with existing weakly supervised methods. However, we agree that the important practical impact of a weakly supervised learning method could be better demonstrated on a larger dataset. To this end, we have been running experiments on the ILSVRC13 ImageNet dataset which has 420K training images, 40K test images, and 200 classes. If the paper is accepted, we'll include the new results on the ImageNet dataset. To the best of our knowledge, we'd be the first to show weakly supervised object detection results on ImageNet.

- The complexity of the algorithm is dominated by the time for computing similarities between the selective search regions. This step is agnostic to the number of classes since we can reuse the precomputed similarity results to compute the coverage score for any class. Thus the algorithm is at most quadratic in the number of regions we generate. This can be improved in various ways, e.g. Lee et al ICCV 2013 and [3] have shown promising results with almost no loss in performance by randomly subsampling the regions. We plan to make available our source code and precomputed region similarities on both PASCAL2007 and ILSVRC13.

- Line 296: we'll clarify the text. Although the set of discovered boxes from the algorithm \Gamma(S) is better localized, it doesn't cover the entire set of positive images. We initially train a detector with the discovered boxes and mine for additional positive examples (one box per each positive image) from images which are not covered. Then we retrain the detector with the union of discovered boxes and mined boxes as the new positive training set. The LSVM refinement process uses this retrained detector as initialization and runs until the latent assignments don't change.

R2:

- We appreciate the insightful baseline suggestion about adding random windows (denoted as "neighboring negative regions" in L366) with less than 50% overlap with respect to the foreground estimates to [27]. Under same experiment conditions as in the second column in Table 2, we added the generated negative regions to the training set as negatives. This actually decreased detection performance for the baseline model [27] from 20.3% (not adding) to 20.2% (adding) and then after latent refinement, the performance dropped from 22.7% (not adding) to 21.8% (adding) on the PASCAL VOC 2007 dataset. This demonstrates that careful generation of mislocalized foreground fragments as hard negatives is critical for accurate detector training, especially if the foreground localization is imprecise. We'll include this baseline comparison.

- We'll include the reference (Deselaers ECCV10, IJCV12). Thank you.

- K is the integer; yes, L136 should have the cardinality.

- Line 254: if the discovered configuration has too few examples (empirically this happens rarely), we use the cluster with the highest purity as a surrogate.

R3:

- It's true that if Delta becomes very large, this factor shrinks. That said, this is a worst-case bound. In practice, greedy algorithms tend to perform better than the worst case. Indeed, we also obtain useful results with this method. In addition, there can be cases where it can still be good to know that the results will not be arbitrarily bad.

- Hough transform noise: While it's true for individual hough transform arrays to have noisy peaks, this phenomenon gets filtered out as we consider all pairwise comparisons among all images with positive labels. Concretely, we find the most frequent edge with cluster labels C_i, C_j, and relative position in the multi-graph G_p = (\mathcal{P}, \mathcal{E}_P) which has a node for each positive image. Algorithmically, this amounts to finding the largest connected component in the multigraph G_p. Fig. 5 shows example discovered co-occurring configurations for all 20 classes in the dataset (e.g. a bicycle with visible wheel, a person's face with torso, etc)

- Line 242: As in [17], we model a visual pattern’s viewpoint with its bounding box aspect ratio (see L309-10). We'll clarify.

- Hard negatives: If a visual pattern configuration is mislocalized (e.g. sofa+lamp in Fig. 5), then a hard negative could incorrectly represent the foreground object-of-interest. With more categories, this will be less of a problem since the context surrounding the object-of-interest will likely be among the set of negative categories.